# Locality matters: Variation in the reproductive cycle and population structure of subtropical sea urchins

Raibel Núñez-González[1]*, Airam N. Sarmiento-Lezcano[2], María J. Caballero[3], Ekin Tilic[4], José Juan Castro-Hernández[1]

**1** Instituto Universitario EcoAqua, Universidad de Las Palmas de Gran Canaria, Las Palmas de Gran Canaria, Spain, **2** Centro Oceanográfico de A. Coruña. Instituto Español de Oceanografía, A. Coruña, Spain, **3** Division of Veterinary Histology and Pathology, Institute for Animal Health and Food Safety (IUSA), Veterinary School, University of Las Palmas de Gran Canaria, Arucas, Spain, **4** Department of Marine Zoology, Senckenberg Research Institute and Natural History Museum, Frankfurt am Main, Germany

\* raibel.nunez@ulpgc.es

## Abstract

The life cycle of many Echinoidea species remains poorly understood despite research conducted in temperate and tropical-subtropical regions. Common species in the Central-Eastern Atlantic's shallow waters include *Paracentrotus lividus*, *Arbacia lixula*, and *Sphaerechinus granularis*. Nevertheless, significant gaps in understanding their life cycles persist. This study discusses the reproductive cycles of three sea urchin species in rocky coastal ecosystems around Gran Canaria Island (Spain) (27º45´N, 15º45´W) from June 2020 to May 2021. Morphological measurements reveal that test length increases without a corresponding weight gain. The average size at first maturity ($L_{50}$) was greater in females (*A. lixula* 46.26 mm; *P. lividus* 46.03 mm; *S. granularis* 49.67 mm SL) than in males (*A. lixula* 43.55 mm; *P. lividus* 42.82 mm; *S. granularis* 48.57 mm SL). The gonadosomatic index in females exceeded that of males. Histological analysis showed oocytes at various developmental stages, indicating asynchronous ovarian development with successive batch spawning seasons. Reproductive activity was generally observed during the warm season for all three species, likely coinciding with increased nutrient availability in Canarian waters. Notably, *P. lividus* was the only species to show two reproductive seasons annually in San Cristobal. DNA analysis confirmed species identification and provided new fragments of the COI gene, now available in GenBank for future population analysis. These findings represent the first reproductive data for these species in the North Atlantic region, offering valuable insights into their populations and establishing baseline information for managing sea urchin populations.

**Data availability statement:** All data generated and analyzed during this study have been deposited in the PANGAEA repository [64], and are accessible via the following DOIs: 10.1594/PANGAEA.983883 [65] and 10.1594/PANGAEA.983888 [66].

**Funding:** The author(s) received no specific funding for this work.

**Competing interests:** The authors have declared that no competing interests exist.

## Introduction

Sea urchins are marine invertebrates that play a crucial role in controlling algae growth by limiting their abundance and distribution, primarily through grazing [1]. However, they are also known to capture suspended particles using their tube feet [2]. The distribution of sea urchins has been associated with some physical and biological factors, such as water motion, food availability, predator presence, and the availability of refuge areas [3,4]. These organisms are dioecious, with no known sexual dimorphism [2]. Regular echinoids possess five gonads distributed in the interambulacral zone, have external fertilization, and indirect development of the larvae [5]. Although the life cycles of many echinoid species remain poorly understood, particularly concerning their geographical distribution and adaptation to specific environmental conditions, research has been conducted on sea urchins in temperate and tropical-subtropical latitudes [6–11]. In the Central-Eastern Atlantic Ocean, *Paracentrotus lividus* (Lamarck 1816), *Arbacia lixula* (Linnaeus 1758), and *Diadema africanum* Rodríguez, Hernández, Clemente & Coppard 2013 are among the most representative species.

The distribution of several sea urchin species has been studied around the Iberian Peninsula, Mediterranean Sea, and Canary Islands [4,12–15]. Due to their high abundance and biomass, *P. lividus*, *A. lixula*, and *Sphaerechinus granularis* (Lamarck 1816) have received particular attention for their role as bioeroders in rocky ecosystems [16–18]. *P. lividus* and *A. lixula* are distributed along the Mediterranean Sea and North Atlantic Ocean, with their special distribution influenced by a sheltered-exposed gradient, hydrodynamic resistance, and microhabitat preference when coexisting [4,19,20]. In the Mediterranean Sea and the Atlantic, *P. lividus* has shown a behavior related to excavating their burrow, which fits the animal's size, and provides protection against their predators and water movement [21]. Meanwhile, *A. lixula* is primarily a grazer of encrusting coralline algae and is typically found in shallow, exposed, and usually vertical rocky areas where it withstands strong water motion and turbulent conditions [4,19,20,22]. In the Canary Archipelago, both species share a similar distribution in shallow rocky reef habitats [19]. On the other hand, it is reported that *S. granularis* is distributed in the Mediterranean Sea to 100 m but is commonly found between 5 and 15 m depth [13,16]. This species inhabits various substrates and plays a significant role in grazing Corallinaceae-dominated areas. In Madeira and the Canary Islands (East Atlantic Ocean), *S. granularis* is found further from the coast (30–40 m) and at depths below 6 m [23,24]. *S. granularis* also displays a covering behavior, often concealing itself with small rocks, algae, or shells [13].

The populations of several sea urchin species have steadily declined in recent years due to overfishing [25–28] and mass mortality events caused by disease outbreaks [29–37]. *P. lividus* is considered an edible species in the Atlantic Ocean and the Mediterranean Sea [38], and in regions such as Galicia (NE Atlantic), where the annual landing has ranged between 400 and 750 tons per year since 1985 [39]. Due to the rising demand for this resource, natural populations are being harvested at higher rates, leading to declines in abundance [39]. *S. granularis* is less commonly

harvested, but with the decline of *P. lividus* populations, it has become an alternative for local consumption in certain regions of the Hellenic Seas [13]. Although there is no official harvesting, data on its population status remains scarce [13]. On the other hand, *A. lixula* is considered a non-edible species and is not directly affected by harvesting. However, some authors suggested that the increasing *A. lixula* populations could impact benthic communities [15]. The loss of these sea urchin populations not only alters the structure of rocky coastal ecosystems but also reduces genetic variability, potentially affecting their adaptability to environmental changes such as temperature fluctuations, ocean acidification, and salinity shifts [40]. As a result, it is expected that natural *P. lividus* populations may require repopulation efforts in the future. Understanding their reproductive cycles is therefore crucial for developing conservation strategies and sustainable fishery management plans [39,41].

The reproductive cycles of these species have been documented in the Northeast Atlantic Ocean and the Mediterranean Sea, and several studies show that the breeding period depends on the locality and the season [6–11]. For example, some authors reported that near the Strait of Gibraltar (Spain), spawning occurs in different seasons depending on each site and is related to the food source in each location [8]. Information about the reproductive cycles of our target species has been reported in previous studies. *A. lixula* shows differences in spawning periods regarding the quantity and season in the Mediterranean Sea [9,10]. The same pattern was reported for *S. granularis* in the Mediterranean Sea and Madeira Archipelago (Central-east Atlantic) [7,11]. In the Canary Archipelago, information on reproduction is scarce or non-existent in species such as *A. lixula* or *S. granularis*, as most research efforts on sea urchins have focused mainly on *Diadema africanum,* a species whose population blooms have been considered an ecological threat with significant consequences for biodiversity and fisheries [42,43].

In this context, the present work aims to contribute to our understanding of the population dynamics of *A. lixula*, *P. lividus*, and *S. granularis,* the most common sea urchin species inhabiting rocky shores around the Gran Canaria Island (Central-east Atlantic, Spain) [14]. Specifically, we investigate their spatial distribution patterns and reproductive cycles across different localities. We hypothesize that these species exhibit reproductive cycle variations in response to the environmental conditions of the Canarian archipelago. To test this, we analyzed their reproductive cycles using a histological examination of oocyte development and reproductive parameters, including length frequency distributions, sex-ratios, length at maturation, and spawning seasons, to determine reproductive strategies. Additionally, DNA barcoding techniques were employed to confirm species identification and provide genetic reference data. These findings are essential for understanding the population dynamics of these species, particularly in the absence of other biological and fisheries-related estimates.

## Materials and methods

### 1. Study area

This study was conducted in Gran Canaria Island, Spain (Central-Eastern Atlantic Ocean), 27°45´N, 15°45´W approximately, in five localities around ("Tasartico" and "La Aldea" in the West, "Arguineguín" in the South, "San Cristobal" in the East, and "Bañaderos" in the North), covering all cardinal points from June 2020 to May 2021 (Fig 1). The map was generated using geographic information system QGIS [44] and bathymetric data obtained from the GEBCO (General Bathymetric Chart of the Oceans) dataset [45].

### 2. Field survey

At least five individuals of each sea urchin species, with different sizes, were manually collected using basic snorkeling equipment, every month during daylight, from 1 to 5 m depth at each locality. The organism collection was performed using a spatula to lever and extract the organisms. Sea urchins of different sizes, from juveniles to adults, were collected to determine the length frequency distribution and provide the length at first maturity. To avoid bias in capture, all

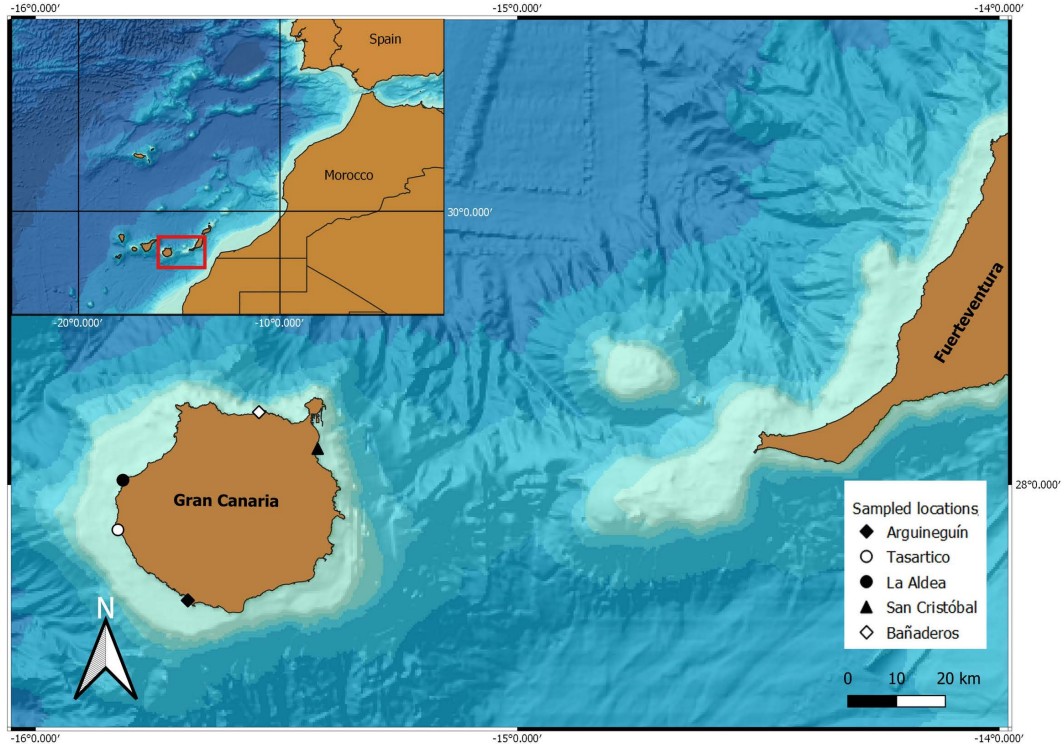

**Fig 1. Sampled locations of sea urchins (*Paracentrotus lividus*, *Arbacia lixula,* and *Sphaerechinus granularis*) around Gran Canary Island, Spain (Central-Eastern Atlantic Ocean).**

the organisms were randomly selected in each area. After this, the animals were transported to the lab in a small case with seawater. In the laboratory, eugenol was placed in a box with seawater to anesthetize them and then euthanasia was performed by placing them in a freezer for a few minutes, to minimize and eliminate pain and distress during this process [46,47].

In addition, subsamples of sea urchins were used (5 individuals of *A. lixula*, 6 *P. lividus*, and 2 specimens of *S. granularis*) for DNA Barcoding analysis to confirm the ID of the organisms in this study. Tissue samples are deposited at the Senckenberg Natural History Museum Frankfurt; the DNA extractions are stored in the Senckenberg DNA Bank (Collection Numbers SMF 6949-SMF 6956). The Government of the Canary Islands allowed the collection of all the sea urchins under the permission Nº: 168/ 2020 – Tomo: 1, Libro: 401, Date: 10/06/2020.

### 3. Laboratory procedures

The sea urchins were weighed (Total weight, TW) with an analytic balance (Brand RADWAG, Model PS 750.R2, precision 0.001 gr), and the test height (oral-aboral axis, H), and test diameter (perpendicular to the oral-aboral axis, D) were measured with a caliper to the nearest 0.01 mm, before the dissection. Both measurements were taken in duplicate to minimize the error. Subsequently, the organisms were dissected using stainless-steel dissection scissors across the test to extract all the internal organs for analysis.

The five gonads and the gut with their content were extracted and weighed (eviscerated weight, eW). Sex was estimated from 644 samples observing the gonads macroscopically following the [6] and [15] descriptions for *A. lixula* and *P. lividus,* respectively. In the case of *S. granularis*, the gonad color is not representative of describing the sex

macroscopically [11]. However, the state of maturity was determined for only 400 individuals through gonad histological analysis, encompassing all maturity stages, following the standard terminology for describing reproductive development [6,11,15]. Gonads were fixed with formaldehyde 4% for histological analysis to confirm the sexes. After fixation, the gonad samples were placed into cassettes and processed following the protocol of [48]. Processing of the samples included dehydration through ascending grades of alcohol, clearing in xylene, and finally imbibition in paraffin wax. Paraffin blocks were sectioned at 4 µm and sections stained with hematoxylin and eosin (H&E). The slides were mounted and examined with a light microscope (Olympus BX51TF, Japan). All the samples were analyzed in the Laboratory of Histology at the Veterinary Faculty, Universidad de Las Palmas de Gran Canaria. Moreover, the gut content was weighed to correlate the Gut Condition Index (GCI) with the reproductive cycle.

## 4. DNA Amplification and Sequencing

In addition, samples of the sea urchin peristomial membrane were taken and stored in absolute ethanol at −20 ºC until processed in Grunelius-Möllgaard Lab (Senckenberg Research Institute and Natural History Museum). Following the supplier's instructions, we extracted total genomic DNA utilizing the DNeasy Tissue Kit (Qiagen). Subsequently, a fragment of the mitochondrial cytochrome c oxidase subunit I (COI) gene (up to 645 bp) was amplified using the specified primers. The HCO2198/LCO1490 primer set by [49] was used for specimens of *A. lixula*. Genes were amplified for *P. lividus* and *S. granularis* specimens using specific echinoderm primers COIe-F/COIe-R [50]. The PCR amplification was performed in a 25 µl reaction mixture containing 21µl of ddH$_2$O, 1µl each of the forward and reverse primers (10µM), and 2µl of extracted DNA and the PureTaq™ Ready-to-Go™ PCR Beads (Cytiva, UK), The PCR program used the COIe-F/COIe-R primers included an initial denaturation at 95 °C/ 2 minutes, followed by 30 amplification cycles (95 °C/30 seconds, 45 °C/30 seconds, and 72 °C/1 minute), and concluded with a final step at 72 °C/8 minutes [51]. The program used for HCO2198/LCO1490 started with initial denaturation at 94°C/3 minutes, 5 amplification cycles (94°C/30 seconds, 47°C/45 seconds, 72°C/ 1 minute), then 30 cycles (94°C/30s – 52°C/45 seconds – 72°C/1 minute), and a final step at 72°C/5 minutes [49]. The reactions were carried out in an Eppendorf thermal cycler. Products that were successfully amplified underwent purification using 2µl of ExoSAP-IT PCR product cleaning reagent. The purified products were sequenced at the Senckenberg Biodiversity and Climate Research Centre (SBiK-F) and assembled with Geneious v.11.0.2 [51].

## 5. Ethics statement

This study involved the collection and analysis of three species of sea urchins, which are not subject to ethical approval under current Spanish legislation. According to "Real Decreto 53/2013", of February 1, which regulates the basic standards for the protection of animals used in scientific research and education in Spain (BOE-A-2013–1337), ethical approval is only required for research involving live cephalopods and vertebrate animals. Therefore, no ethical approval was necessary for this study. In addition, no endangered or protected species were involved, and all samples were collected following local environmental regulations. All procedures complied with institutional, national, and EU regulations for the ethical treatment of marine invertebrates.

## 6. Data analysis

Relationships between test diameter (D), test height (H), and total wet weight (TW) of the three species were obtained to describe the population growth structure around the island. Regression of D vs. H was performed, and then a Standardized Major Axis Regression (SMA) [52] was performed using *smatr* package in R Studio [53] to determine if there is substantial evidence to reject this $H_0$ that the regression slopes between localities are equal. The regression values were then compared across localities using ANCOVA [54], performed by the *emmeans* package in R Studio [55]. The allometric growth of TW concerning the test diameter was estimated using a potential relationship $TW = a \cdot D^b$, where $a$, is the intercept of the regression model, and $b$ is the slope (or allometric coefficient). For TW versus D, the allometric coefficient

of 3 indicates isometry, i.e., that sea urchins' wet weight and diameter have a proportional growth. If the growth model is negatively allometric, the body length grows faster than the body weight. Differences between the expected value from isometric growth and values of the regression coefficient (b) were compared using a t-test [56]. This test evaluated the null hypothesis $H_0$: b = 3 in the TW *vs.* D relationship with a significance level of 5% (α = 0.05) and a critical value of t0.05, > 200 = 1.98 [56,57].

The sex ratio was calculated considering the histologically verified sexes, and to determine if the variables were independent using a Chi-square test performed with *stats* package in R Studio [58]. Then, a Fisher test was performed with *stats* package in R Studio [58], to determine if there was sex dominance across the localities. Also, to compare the size of the oocytes between species, the diameter of mature oocytes of 30 adult females (10 of each species, and 10 eggs per individual) was measured, using slides with histological images, and the cell measures were made using ImageJ software [59]. After the measurement, an ANOVA and a Tukey test were performed with *stats* package in R Studio [58] to compare the size of the oocytes between species. All the statistical analysis was performed using R Studio software [58].

For mature individuals (n = 187), maturity ogives, the length at first maturity ($L_{50}$) for both sexes, and the percentage of mature individuals by length class were calculated. The data were fitted to a normal cumulative distribution using an iterative nonlinear regression model. Additionally, a comparison of means test was conducted to assess potential differences between the $L_{50}$ values of both sexes. The resulting data were modeled using a sigmoid function as described below:

$$P_r = \frac{100}{1 + e^{-r(L-L_{50})}}$$

where $P_r$ represents the percentage of sexually mature individuals, $r$ is a constant indicating the slope of the curve, $L_{50}$ denotes the length at which 50% of individuals were mature, and $L$ refers to the sea urchin diameter at which the $P_r$ was calculated.

The gonadosomatic index (GSI) was calculated using the ratio of the wet weight of the five gonads to the total wet body weight of the individual to determine the spawning season (Equation 1). The Gut Condition Index (GCI) was calculated using the ratio of the wet weight of the stomach content to the total wet body weight of the individual, calculated following [60] (Equation 2).

$$GSI = \frac{Gonad\ weight\ (gr)}{Total\ weight(gr)} \tag{1}$$

$$GCI = \frac{Stomach\ weight\ (gr)}{Total\ weight(gr)} \tag{2}$$

After the GSI analysis, ANOVA and a Tukey test were performed on R Studio [58] to compare the GSI values between species and localities. Moreover, an alternative approach to determine the spawning season involved analyzing the temporal variation in the frequency distribution of individuals across different stages of maturity, based on macroscopic observations of the gonads. The number of sea urchins in each maturity stage and their corresponding frequencies were computed. To estimate the Gonadosomatic Index at 50% maturity ($GSI_{50}$), a logistic function was applied to the proportion of mature sea urchins across 0.5 GSI intervals for both sexes, using a nonlinear least-squares regression method. The logistic equation was:

$$PM_{GSI} = \frac{100}{1 + e^{-a(GSI-b)}}$$

where $PM_{GSI}$ is the percentage mature at GSI, $a$ is the slope, and $b$ is the $GSI_{50}$. The minimum size at maturity of males and females was taken as the smallest specimen with a GSI over $GSI_{50}$ based on the GSI–SL relationships [60].

In addition, *dplyr* [61] and *tidyr* [62] were used for data manipulation, and *ggplot2*package [63] was used for data visualization.

All data generated and analyzed during this study have been deposited in the PANGAEA repository [64], and are accessible via the following DOIs: 10.1594/PANGAEA.983883 [65] and 10.1594/PANGAEA.983888 [66].

## Results

### 1. General characteristics of the sea urchin populations

During the sampling year, 677 sea urchins (252 of *A. lixula*, 359 of *P. lividus*, and 66 of *S. granularis*) were collected in the five localities described before (Fig 1), with a high frequency of organisms in the class size between 30–50 mm (Fig 2). Body measurements of sea urchins (Table 1) showed a variation in diameter for the species examined (*A. lixula*, 9.54 to 62.1 mm; *P. lividus*, 10.8 to 64.5 mm; *S. granularis*, 25.1 to 68 mm), where the smallest species was *P. lividus* (Table 1). *S. granularis* was the largest species (considering diameter means and height), followed by *A. lixula* and *P. lividus*. The normality of the test diameter data was assessed using the Shapiro-Wilk test, which indicated that the data were significantly non-normal, and a significant difference in diameter was observed for *A. lixula* and *P. lividus*, between the localities (Shapiro-Wilk test, $F_{2,300} = 36.85$, $p < 0.005$), where the Tukey test showed that Tasartico was the locality with the smallest organisms of both species of the sampling areas ($p < 0.005$).

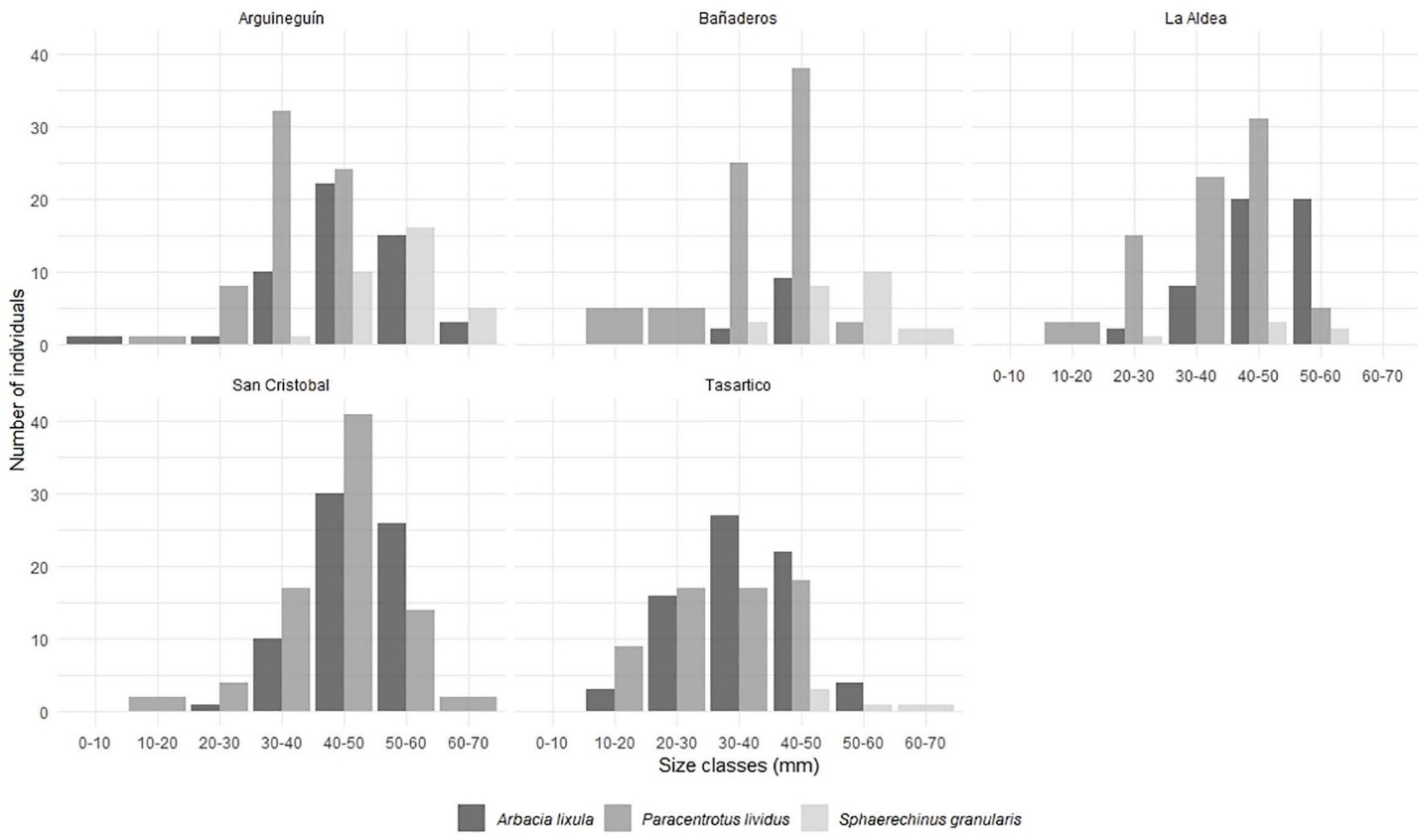

**Fig 2. Distribution of sizes across the localities, for each sea urchin species (*Paracentrotus lividus, Arbacia lixula,* and *Sphaerechinus granularis*).**

**Table 1. Body characteristics and the number of individuals sampled by species and locality.**

| Locality | Species | Diameter (mm) | Height (mm) | Total weight (gr) | Gutted weight (gr) | n |
|---|---|---|---|---|---|---|
| Bañaderos (North) | A. lixula | 42.43±4.27 | 19.37±4.27 | 36.60±10.76 | 23.05±5.80 | 11 |
| | P. lividus | 38.75±8.73 | 18.52±4.91 | 31.01±15.05 | 17.83±7.66 | 76 |
| | S. granularis | 49.42±8.73 | 29.22±5.95 | 65.59±29.02 | 37.31±15.71 | 23 |
| San Cristóbal (East) | A. lixula | 46.78±7.01 | 20.66±3.50 | 46.92±17.38 | 29.98±9.57 | 67 |
| | P. lividus | 43.44±8.62 | 21.09±4.93 | 40.58±16.85 | 23.72±9.25 | 80 |
| | S. granularis | – | – | – | – | 0 |
| Arguine-guín (South) | A. lixula | 46.29±9.52 | 20.78±5.22 | 47.16±21.23 | 28.47±10.00 | 52 |
| | P. lividus | 37.76±7.43 | 18.05±4.79 | 28.07±14.25 | 16.64±7.18 | 65 |
| | S. granularis | 53.83±6.23 | 32.64±4.57 | 75.53±30.06 | 42.11±15.44 | 32 |
| Tasartico (West) | A. lixula | 35.71±9.36 | 15.46±4.28 | 24.55±15.36 | 16.45±9.05 | 72 |
| | P. lividus | 32.06±9.71 | 15.31±4.91 | 20.13±13.75 | 11.72±7.23 | 61 |
| | S. granularis | 52.96±8.48 | 33.62±5.13 | 76.32±35.51 | 40.92±17.31 | 5 |
| La Aldea (West) | A. lixula | 45.80±7.87 | 20.12±4.06 | 44.45±16.73 | 27.52±9.58 | 50 |
| | P. lividus | 37.02±9.10 | 17.74±5.14 | 28.83±16.61 | 17.45±8.81 | 77 |
| | S. granularis | 44.79±10.34 | 20.27±4.41 | 42.69±18.52 | 27.73±11.50 | 6 |

The regression between diameter and height was positive and significant ($P < 0.05$) in all the species (Fig 3), SMA results show that the relation between D and H varied across localities just in *A. lixula* (SMA, $F_{4,171} = 14.45$, $p = 0.005981$) and *P. lividus*: (SMA, $F_{4,175} = 9.929$, $p = 0.005981$), and there was insufficient evidence to reject the $H_0$ among *S. granularis* samples differ across localities (SMA, $F_{3,54} = 6.094$, $p = 0.1071$). ANCOVA shows that Tasartico was statistically different from the other localities in *P. lividus* (ANCOVA, $F_{9,349} = 211.2$, $p = 2.2 \times 10^{-16}$) and *A. lixula* (ANCOVA, $F_{9,242} = 154.4$, $p = 2.2 \times 10^{-16}$).

We applied the following allometric growth test diameters-to-total weight (Fig 4) equation to determine relationships in the individuals measured for each species (*A. lixula*: TW = 0.002 $D^{2.649}$; $R^2 = 0.88$; $p < 0.001$; N = 252; *P. lividus*: TW = 0.002 $D^{2.682}$; $R^2 = 0.91$; $p < 0.001$; N = 359; *S. granularis*: TW = 0.001 $D^{2.778}$; $R^2 = 0.85$; $p < 0.001$; N = 66). We observed a significant relationship between the average test diameters and total weight (Fig 4). The t-test showed that the null hypothesis of equality in the regression coefficient $H_0$: b = 3 was rejected for all individuals showing negative allometric growth (t-test, *A. lixula:* t = −10.05 < $t_{0.05,251}$ = 1.96, $p = 0.0001$; 95% Coefficient of Length (CL): 2.58–2.72; *P. lividus:* t = −16.99 < $t_{0.05,358}$ = 1.96, $p = 0.0001$; 95% CL: 2.64–2.72; *S. granularis:* t = −2.53 < $t_{0.05,65}$ = 1.99, $p = 0.0001$; 95% CL: 2.60–2.95).

## 2. Reproductive cycle

**2.1. Histological analysis.** A total of 353 gonad samples of sea urchins across five localities were analyzed. The sex ratio of each species was not statistically different from 1:1 (Chi-square test: *A. lixula*, 1:1.2 $X_0^2 = 1.3158$; N = 171; $p > 0.05$; *P. lividus*, 1:1.1 $X_0^2 = 0.6914$; N = 175; $p > 0.05$; *S. granularis*, 1:0.7 $X_0^2 = 1.8519$; N = 54; $p > 0.05$). The Fisher test shows that the differences between sexes across the localities were not statistically significant ($p = 1$; N = 353). Macroscopic observations were validated with gonad tissue histology for each species (Figs 5- 7). To describe gametogenesis and document the annual reproductive cycle, the predominant cell populations in the germinal epithelium were examined histologically (Fig 5 (A. lixula), Fig 6 (P. lividus), Fig 7 (S. granularis)). The gonadal growth pattern was divided into five stages, following descriptions by [67,68].

***Stage I: Gonad growing.*** The first stage of maturity is characterized by the presence of large nuclei near the base of the germinal epithelium belonging to primitive spermatogonia or oogonia. In some cases, we could find residual oocytes or spermatozoa from the previous spawning event.

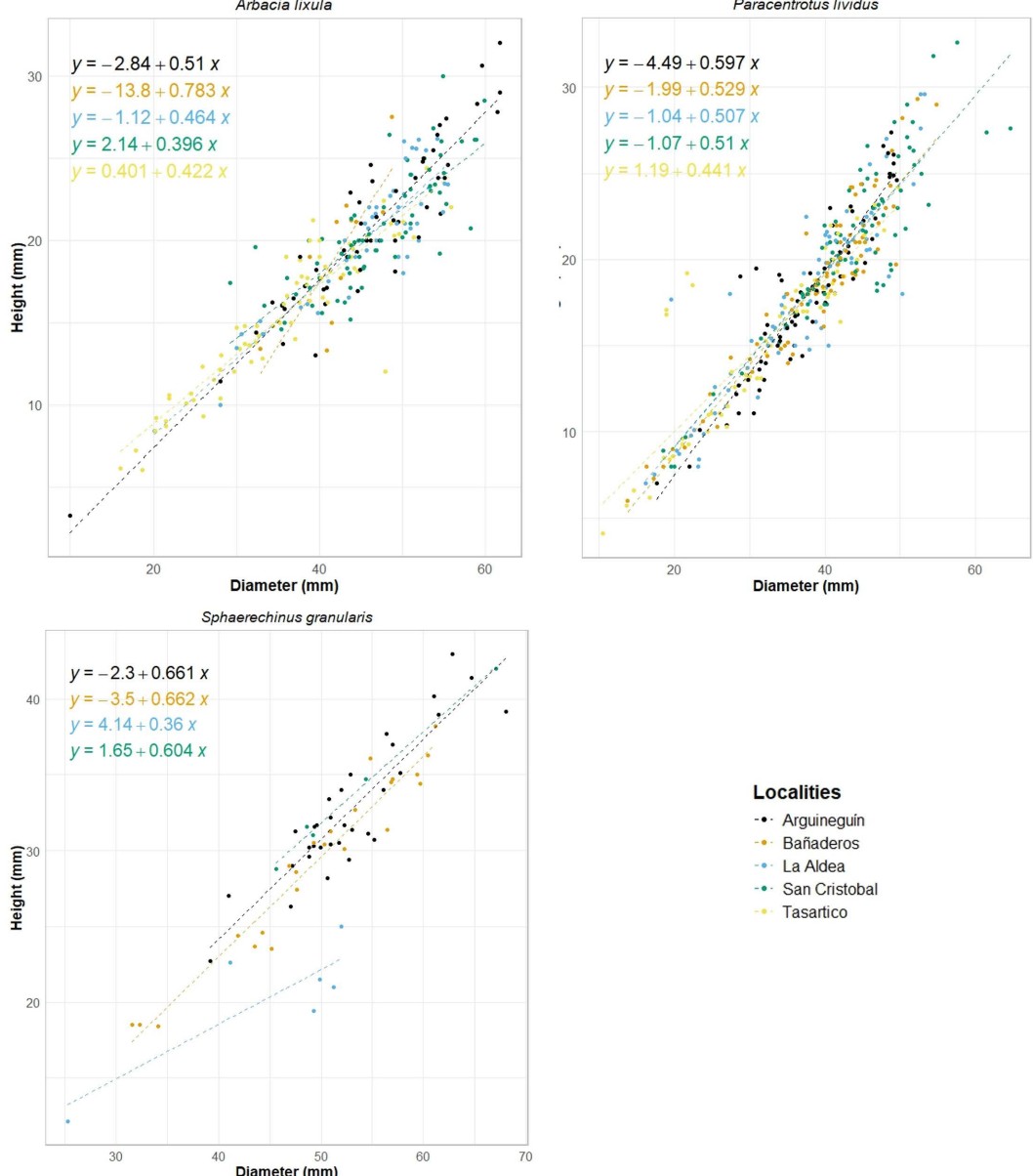

**Fig 3. Relationships between diameter and height of the *Paracentrotus lividus*, *Arbacia lixula*, and *Sphaerechinus granularis*, considering all the localities.**

*Stage II: Premature stage.* This stage is characterized by mature cells, the presence of nucleated cell remnants, and thickened nutritive tissue at the edge of the ovaries or testes. In the case of the ovaries, it was common to find oocytes in different stages of maturity.

*Stage III: Mature stage.* This stage occurs when there is a higher proportion of densely packed mature cells (oocytes or sperm) and the nutritive tissue has considerably decreased in thickness, sometimes becoming indistinguishable.

*Stage IV: Spawning.* This stage is represented by large acini containing empty spaces, with numerous unspawned oocytes or spermatozoa. The acinar wall is almost devoid of any cells.

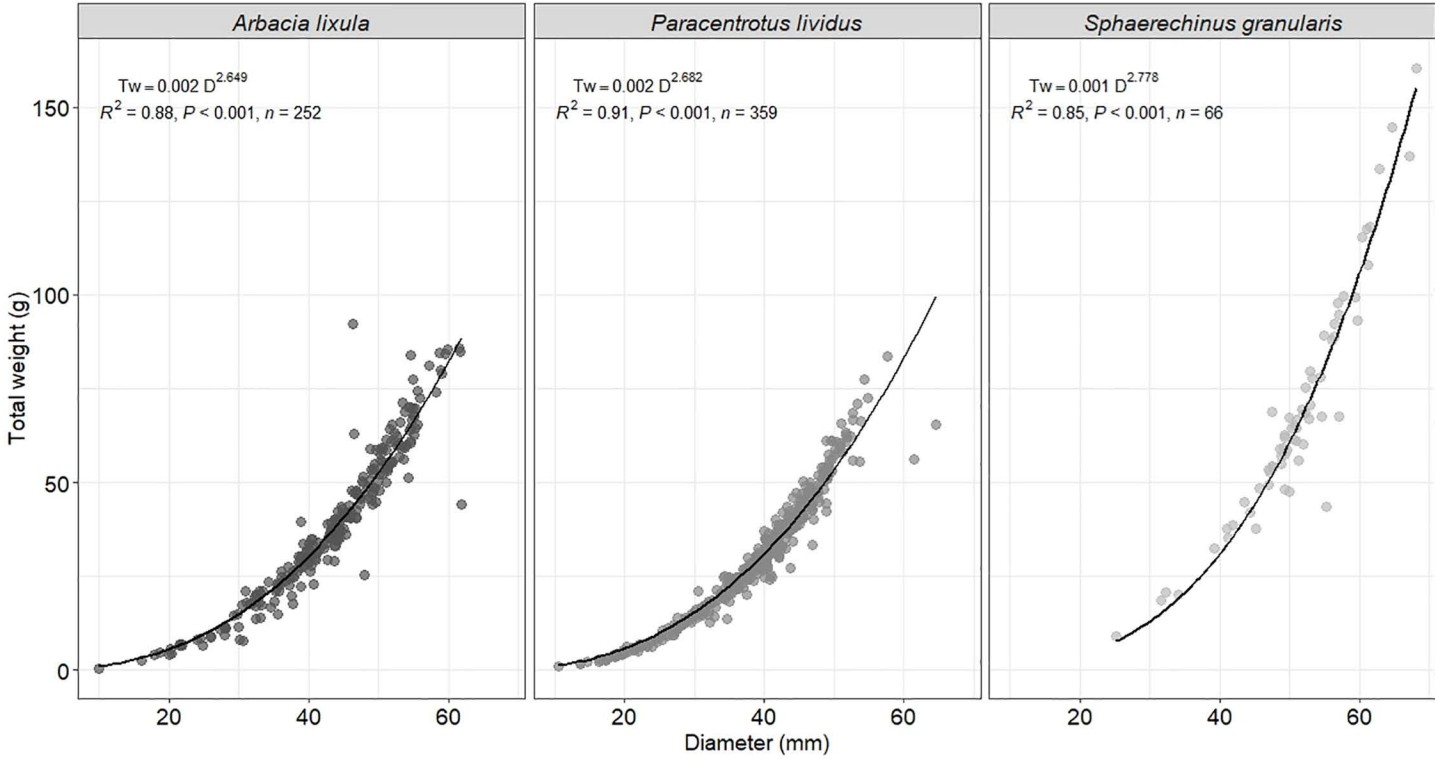

**Fig 4. Relationships between diameter and the total weight of the *Paracentrotus lividus*, *Arbacia lixula*, and *Sphaerechinus granularis*, considering all the localities.**

***Stage V: Recovery.*** At this stage, the gonadal tissue is mostly depleted of reproductive cells, with remnants of mature sperm or oocytes. In some cases, the beginning of the appearance of germinal epithelium and thickening of the acinar wall can be seen.

Additionally, a significant difference in oocyte diameter during the III stage (Mature) between the 3 species (ANOVA, $F_{2,300} = 36.85$, $P < 0.001$) was found. The post hoc test showed that the difference was between all the species (all the values of $P < 0.001$), being the oocytes of *A. lixula* smallest in size, and those of *S. granularis* being the biggest in comparison to the other two species (Table 2).

**2.2. Frequency of reproductive stages, length at first maturity, and gonadosomatic index.** In Fig 8, the reproductive stage frequency shows that both sexes mature synchronously. Different maturity stages were recorded each month after observing the gonad maturity of the species examined in this study (see section 3.2.1). The seasonal distribution of the five stages showed that stage II (premature) and stage III (mature) were present for almost the whole year for both sexes and in the 3 species. In *A. lixula*, a high frequency of immature (stage I) was recorded between January and February, stage II (premature) during March and May, stage III (mature) between June and August, stage IV between October and December, in all the localities, and both sexes. However, in *P. lividus* the frequency of each stage did not follow the same pattern across the localities: the highest percentage of Stage I in females appears between June and August, but in males the pattern changes with high percentages during March and June in Arguineguín and Bañaderos, just in June in El Puertito, from April to June in San Cristóbal and just in April in Tasartico. In the case of stage III, in Bañaderos and El Puertito, females and males were mature in different months across the year (females in February, and males in August and November; and females in July and males in April, respectively); in San Cristóbal,

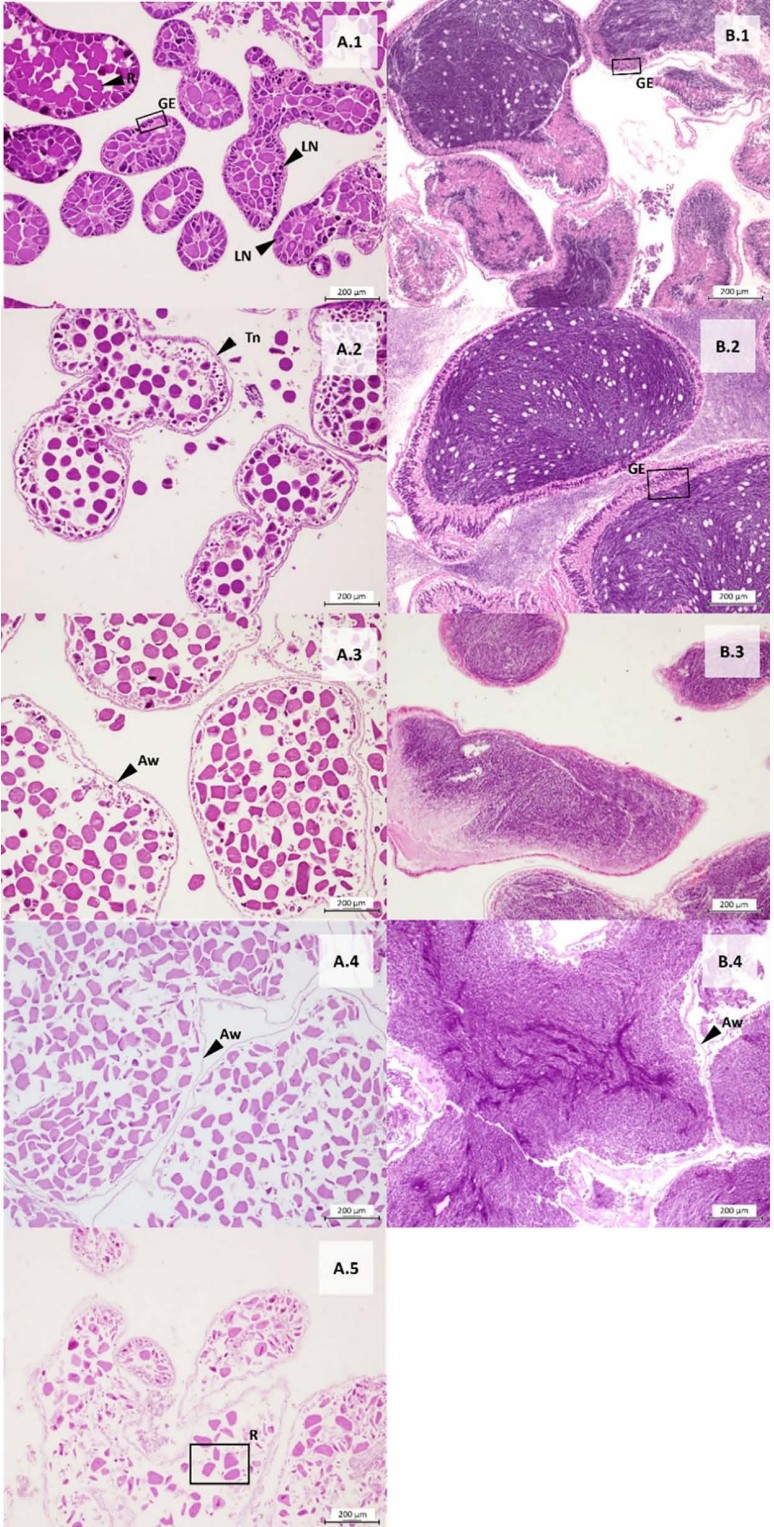

**Fig 5. Histological sections of female and male gonads of *Arbacia lixula*.** (A) Cross section of the ovary, and (B) Cross section of testicles. The number indicates stages: 1-Gonad growing, 2-Premature, 3-Mature, 4-Spawned, and 5-Recovery. LN: Large nuclei cell, GE: Germinal epithelium, Tn: Thickened nutritive tissue, Aw: Acinar wall devoid of cells, R: remnant of mature cells. (Scale bar = 200 μm).

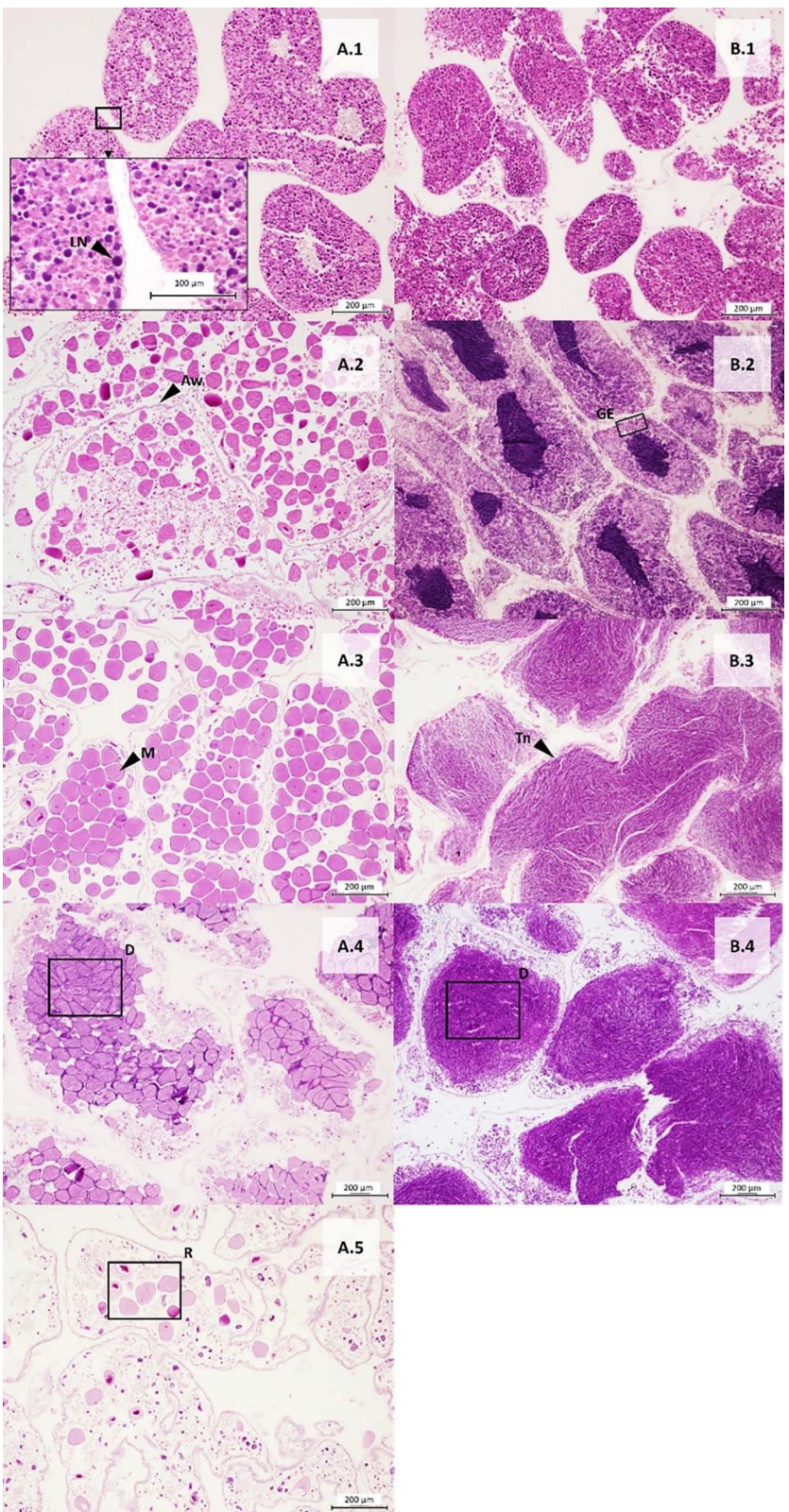

**Fig 6. Histological sections of female and male gonads of *Paracentrotus lividus*.** (A) Cross section of ovary, and (B) Cross section of testicles. The number indicates stages: 1-Gonad growing, 2-Premature, 3-Mature, 4-Spawned, and 5-Recovery. LN: Large nuclei cell, GE: Germinal epithelium, Aw: Acinar wall devoid of cells, D: Densely packed mature cells, M: Mature oocyte, R: remnant of mature cells. (Scale bar = 200 μm).

females and males are mature during August, and in Tasartico during September. Considering Stage IV, in Arguineguín we found spawned female organisms from September to February and males from December to February. In the case of Arguineguín and Bañaderos show a similar pattern, with a high frequency of female organisms in Stage IV from September to November, and males between December and February; in El Puertito, que found females in this stage from November to February; in San Cristobal, females and males show a similar pattern with a high frequency of individuals in this stage between February and March; and Tasartico females were in this stage between September and January, and males between December and March.

Finally, *S. granularis* shows that there were organisms in Stage I across the year in all the localities, but with a high frequency of females in January and males in February and May at Tasartico, females in July, and Males in January and May at Arguineguín, and both sexes in May in Bañaderos. Stage II was not present in the sampled organisms. In the case of Bañaderos, Stage III was present during March and August in both sexes, and in Arguineguín both sexes in different months across the year (females: August, October, February, and May; and males: June, September, October, and January). The last, Stage V, was found in samples of Tasartico, with a high frequency of organisms during January in females, and February and May in males.

The length at first maturity ($L_{50}$) of sea urchins around Gran Canaria Island showed that *A. lixula* reached the $L_{50}$ at 43.55 mm in males and 45.26 mm in females (Fig 9). $L_{50}$ was estimated for *P. lividus* in 42.82 mm for males and 46.03 mm for females, while for *S. granularis*, $L_{50}$ values were estimated to be 48.57 mm and 49.67 mm for males and females, respectively.

The GSI generally varied between species, localities, and months as expected (Fig 10). *A. lixula* showed the highest values of GSI in October in San Cristobal and Tasartico, in August in El Puertito and Arguineguín, and in February in Bañaderos. The values of GSI for *P. lividus* showed two peaks during March and October/November (Bañaderos, San Cristóbal, and Arguineguín). However, for the rest of the localities, we observed one reproductive season in *P. lividus* (November in Tasartico; May to July in El Puertito). During the year sampled, *S. granularis was* better represented in "Bañaderos" and "Arguineguín" localities, with higher values of GSI during the summer season (August) (Fig 10). We found a significant difference in GSI values in *A. lixula* and *P. lividus*, between localities ($F_{7,677} = 3.186$, $p < 0.05$). The post hoc test (Tukey test) showed that *A. lixula* GSI values were different in "San Cristóbal" than "Tasartico" and "Arguineguín", and for *P. lividus* GSI values were different in "San Cristóbal" than the rest of the localities, and between "Tasartico" and "La Aldea" (all the values of $p < 0.05$).

Based on the logistic function, the global $GSI_{50}$ values were 6.04, 4.67, and 4.34 for *A. lixula, P. lividus, and S. granularis*, respectively (Fig 11). For males and females, the $GSI_{50}$ for *A. lixula* was 4.24 and 5.21, for *P. lividus* was 1.74 and 3.61, and *S. granularis* was 3.08 and 3.91, respectively.

### 3. DNA barcoding analysis

We obtained approximately 663 bp of COI sequencing, 5 individuals of *P. lividus*, and 2 individuals of *S. granularis*, except for 1 individual of *A. lixula* from the Gran Canaria populations. All sequences have been deposited in the GenBank database (Access numbers PQ722060-PQ722067).

## Discussion

### 1. General characteristics of the sea urchin populations

According to morphometric measurements, the maximum diameter recorded for *A. lixula* (62.10 mm) in our study is comparable to other localities such as the Algerian west coast, Mediterranean Sea (63.10 mm) [10], Pagasitikos Gulf, Greece

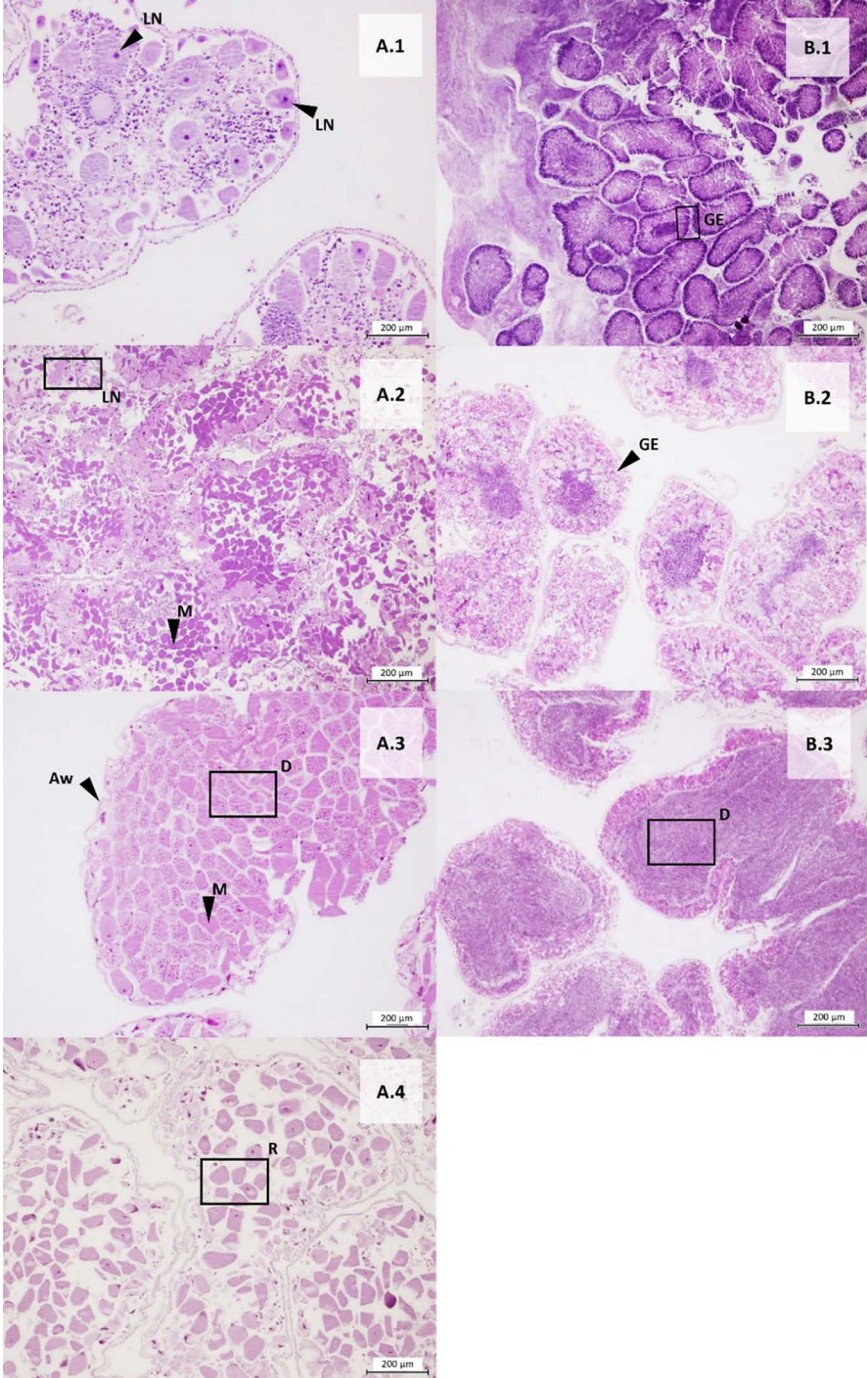

**Fig 7. Histological sections of female and male gonads of *Sphaerechinus granularis.*** (A) Cross section of ovary, and (B) Cross section of testicles. The number indicates stages: 1-Gonad growing, 2-Premature, 3-Mature, 4-Spawned, and 5-Recovery. LN: Large nuclei cell, GE: Germinal epithelium, Tn: Thickened nutritive tissue, Aw: Acinar wall devoid of cells, D: Densely packed mature cells, M: Mature oocyte, R: remnant of mature cells. (Scale bar = 200 μm).

Table 2. Mature oocyte sizes of the 3 species (*Paracentrotus lividus*, *Arbacia lixula,* and *Sphaerechinus granularis*). N = 100 per species.

| Species | Minimum size (µm) | Maximum size (µm) | Mean ± standard deviation |
|---|---|---|---|
| *Arbacia lixula* | 34.76 | 70.80 | 53.20 ± 6.66 |
| *Paracentrotus lividus* | 48.66 | 84.37 | 62.88 ± 7.50 |
| *Sphaerechinus granularis* | 41.19 | 87.83 | 59.05 ± 9.42 |

(54.66 mm) [15]. In the case of *P. lividus* the maximum diameter (64.50 mm) was similar to other studies, such as the one reported by [69] with a maximum diameter of 69.5 mm, on the Moroccan coast; in the case of *S. granularis* the maximum diameter (68 mm) was smaller than the reported in other areas such as Madeira, Portugal (90 mm) [11].

In addition, it is important to highlight that we found significant differences between the localities in test diameter just in *A. lixula* and *P. lividus*, and the locality different from the others was Tasartico, with the smaller organisms. Additionally, the relationship between the diameter and height followed a similar pattern across all three species, showing a significant positive relation between both variables. Regarding morphometric analysis, we found significant negative allometric relationships, which coincide with the research of [15] with *A. lixula* and [70] with *P. lividus*. The high standard deviation values in total weight can be explained by the fact that the sea urchins were in different stages of maturity each month during the sampling campaigns.

Although the 3 species share the same microhabitat in the sampling sites, some works explain that these species can use different resources, even in barren habitats with severe food shortages. For example, [71] reports that *A. lixula* consumes principally encrusting coralline algae (more than 70%), while *P. lividus* prefers fleshy algae (more than 60%). In the case of *S. granularis*, it is a species that consumes mostly encrusted algae [16], and in some cases can consume dead leaves of *Posidonia oceanica* with their epiphyte organisms when they are accessible [72]. Also, the characteristics of each site regarding the current, temperature, water movement, food availability, and anthropogenic influence may affect the presence of each species along Gran Canaria Island. Considering this, the low density of *S. granularis* close to the coast could be explained because of the high effect of water movement in all the localities, and the collection method used. [24] described this species as well distributed in calm areas as deeper zones on Madeira Island. This explains why we found them easily in the quietest areas such as the natural pools of Bañaderos and Arguineguín.

## 2. Reproductive cycle

**2.1. Histological analysis.** The histological analysis of female gonads of sea urchin species showed oocytes in different developmental states, indicating that this species has asynchronous ovarian development with successive batch spawner seasons, as observed in *P. lividus* from other latitudes [6,69,73]. In addition, in Gran Canaria Island, males and females of *P. lividus* mature simultaneously, which agrees with the results of [60] who worked with the same species on the Atlantic coasts of France.

**2.2. Frequency of reproductive stages, length at first maturity, and gonadosomatic index.** Our results of reproductive seasons in *A. lixula* showed a variation according to the localities, mainly concentrated in autumn (in Bañaderos, San Cristobal, and Arguineguín in September, in Tasartico and La Aldea in November). This result differs from the report of [9] with a reproductive season between May and July in the Mediterranean Sea. Furthermore, in our study, we highlighted 2 reproductive peaks (with GSI values higher than the GCI) only in *A. lixula* in San Cristóbal and Tasartico, and *P. lividus* in San Cristóbal, with the highest values between summer and autumn, and the second and lowest between winter and spring (Fig 9). Recently, [10] also found two spawning periods for *A. lixula*: massive between spring and summer and less intense during autumn in the southwestern Mediterranean. In the case of *A. lixula*, GSI values were higher when the sea temperature was higher too, showing the strong relationship between both variables [20].

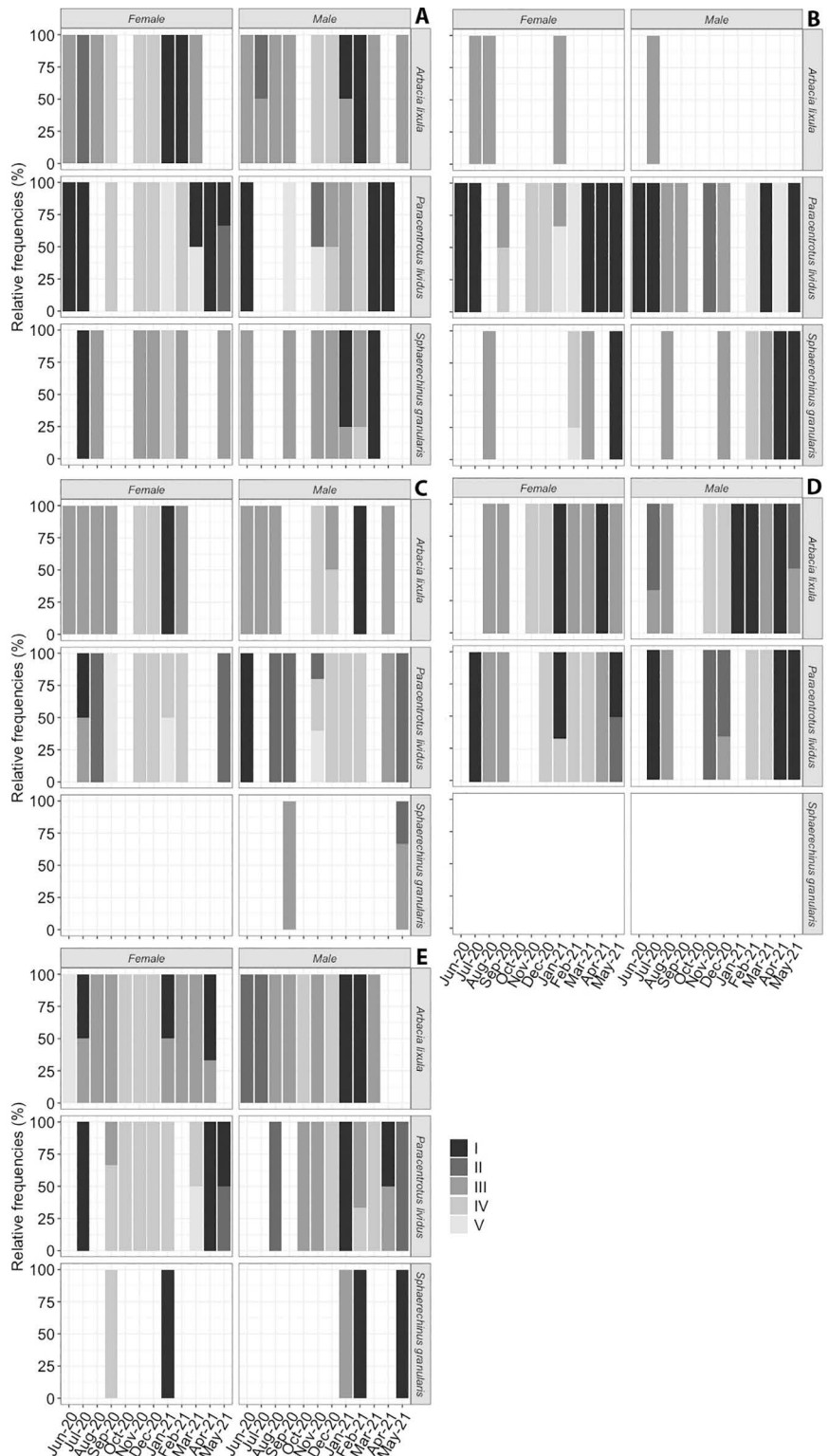

**Fig 8. Relative frequency of the reproduction stage along the year per locality.** (A) Arguineguín, (B) Bañaderos, (C) La Aldea, (D) San Cristóbal, and (E) Tasartico, by sex and species. I) Gonad growing; II) Premature; III) Mature; IV) Spawned; V) Recovery.

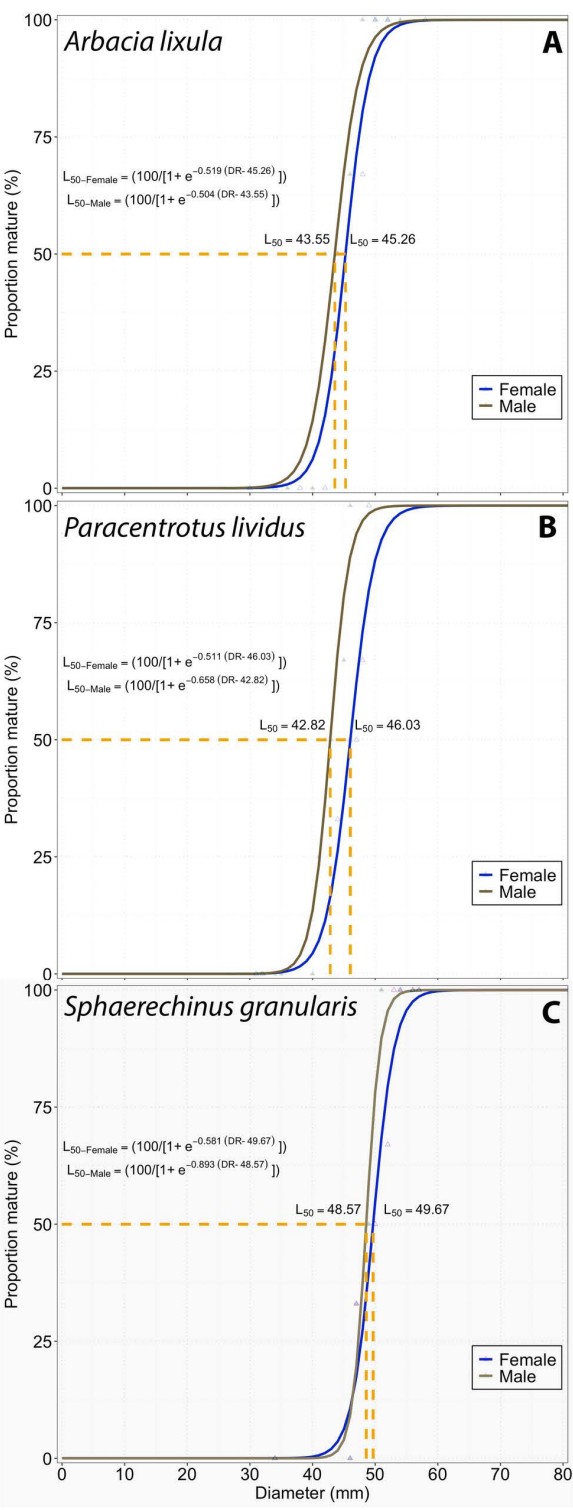

**Fig 9. Relationship between the diameter (mm) and the proportion of mature organisms by sex, of the three sea urchin species.**

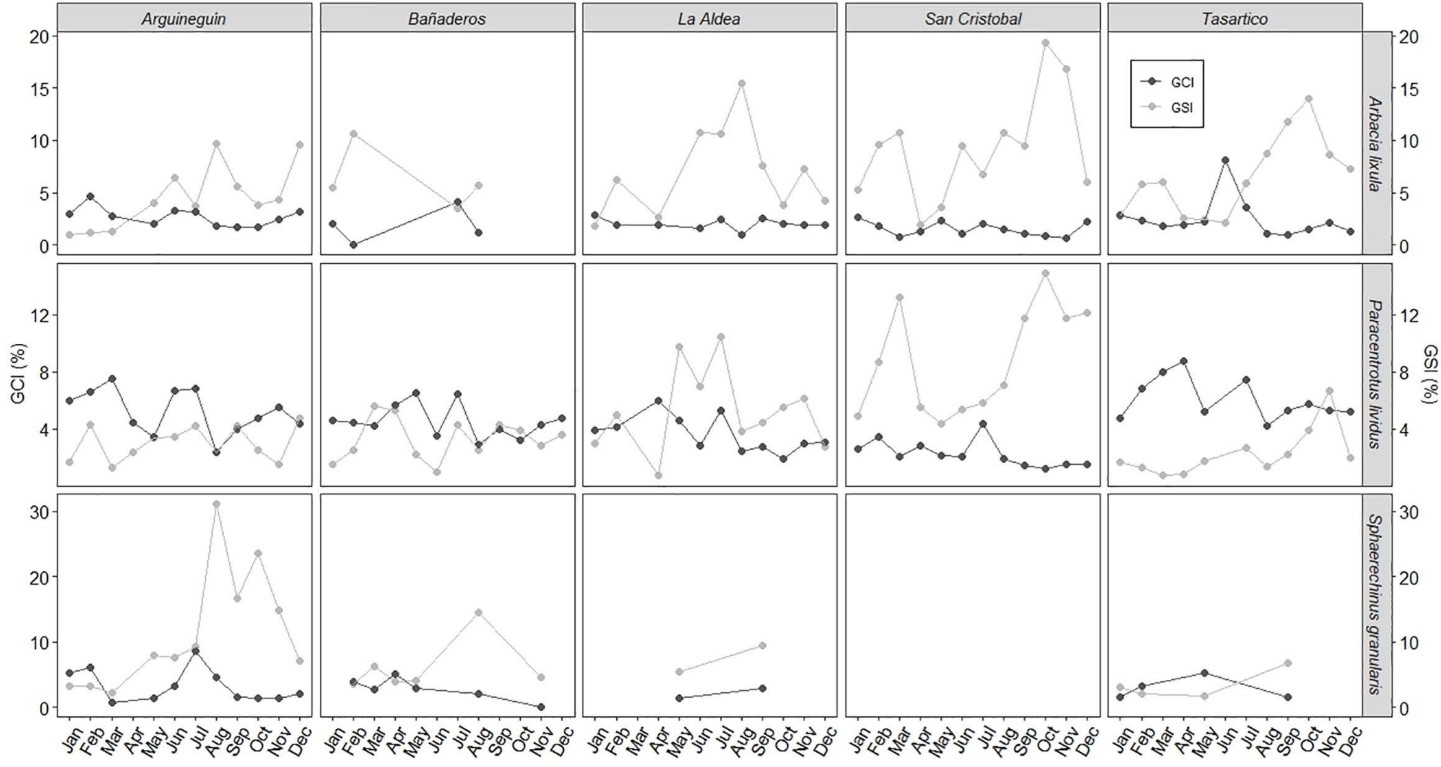

**Fig 10. Gonadosomatic Index (GSI) and Gut Condition Index (GCI), in each locality by species along a year of sampling.**

Our study demonstrates that the reproductive cycle of *P. lividus* varies depending on the seasons and locations. The maturity stage (III) aligns with late summer to early autumn, consistent with findings in the Cantabrian Sea [73]. Regional differences include late spring/early summer maturation on the French coast [60], and the peak of spawning in spring to early summer in the northeastern Spanish Mediterranean [74]. In Sardinia, *P. lividus* shows a primary spring spawning period [75], similar to our observations.

Considering the Canary Archipelago, [76] reported that *P. lividus* shows the main reproductive peak during August, October, and December; [77] reported several maturity seasons throughout the year in Tenerife Island, although the main period was between September and February. The maximum values of GSI were reported during summer, which coincides with our results in Gran Canaria.

In contrast, [75] reported two spawning events for *P. lividus* in a Marine Protected Area of Sardinia (June and March) with less anthropogenic pressure, suggesting a successful gamete production compared to a high-pressure zone, where only one spawning event occurred, and fertile individuals were less abundant. Smaller individuals in high-pressure zones matured earlier, while larger individuals in low-pressure zones contributed more to reproduction due to their capacity for larger gonads and higher gamete output. Similarly, in Bañaderos and Arguineguín, high anthropogenic pressure was associated with lower GSI values.

In the case of *S. granularis*, [11] reported that the GSI had higher values in November, with two spawning peaks: one stronger in November and one weaker in April, in Madeira Island. However, [13] report that the major spawning event of *S. granularis* in the Aegean Sea occurs in spring, and then (with no histological corroboration) a second minor spawning event in early autumn (September), and [78] report two spawning seasons in the Algerian coasts: first during spring

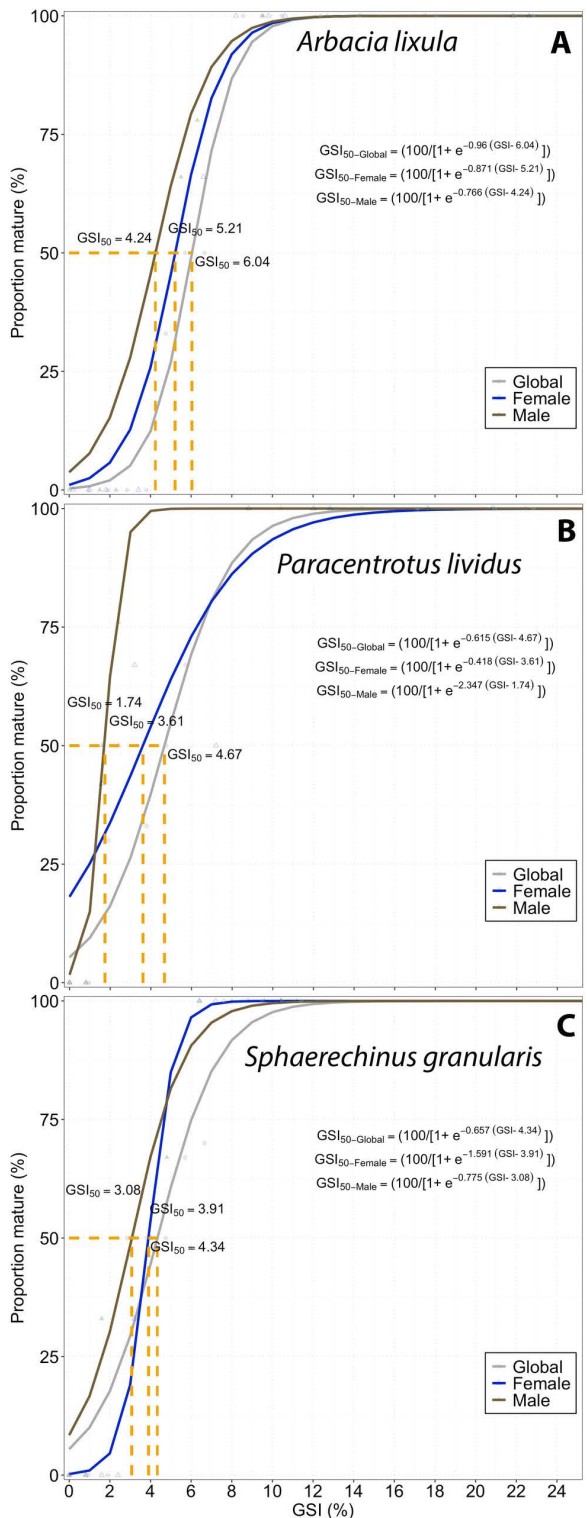

**Fig 11. Relationship between the gonadosomatic index (GSI) and percentage of mature individuals, by sex of the three sea urchin species.**

and the highest peak in autumn. In contrast, [79] in Western Brittany shows that a reproductive peak was marked by a decrease in the GCI and an increase in the water temperature during summer (June to August). Our results are consistent with those described by [79] and show higher GSI values in all the sampling sites from August to September.

One explanation we found regarding the differences in the reproductive cycle among localities is the process explained by [6], which shows that the differences in food availability to the population's habitats may lead to a variation in their nutritional conditions. In addition, [80] research on *Echinus esculentus* on Scotland's coasts, provides information about how during winter months when the nutrient concentration increases in the water column induces a rapid development of gonadal mass, consequently, the quality and availability of the nutrients affect the GSI, and then cause the reproductive cycle to vary.

Another explanation was suggested by [81], who hypothesized that spawning in *P. lividus* is linked to seasonal phytoplankton blooms. In the case of other species of sea urchin, *Echinometra lucunter*, in the Caribbean Sea, the rise of the temperature stimulates the gonad development and the increase of the phytoplankton because the coastal upwelling triggers partial or full spawning in both sexes at the same time [82]. In the Canary Archipelago, several oceanographic processes affect phytoplankton development, such as the seasonal variation of the currents, the eddies, and the wind. According to the research of [83], the maximum wind speed in Gran Canaria Island was during the summer season, and this pattern is highly related to the presence of the eddies, which seems to increase the primary production because of the upwelling of deep nutrient-rich water into the euphotic layer in oligotrophic waters. In the case of the southwest of Gran Canaria, during the summer season, specifically in June, there is a prevalence of cyclonic eddies, with an increase of nutrients causing an increase of the phytoplankton. Additionally, they reported that the phytoplankton increased during autumn or the beginning of the winter season. Our results show that the peak of the reproductive season in almost all the areas around Gran Canaria Island was related to these two seasons: high during summer and lower during autumn, which could be closely associated with spawning in the season where sea urchin larvae can find high concentrations of food for their development.

There is scarce information about the sexual maturity of these species in the Central-Eastern Atlantic, specifically in the Canary Islands area. This knowledge gap highlights the importance of our results, collected over one year in five different localities for the three most abundant species in the area. In this sense, [84] and [85] described the sexual maturity of *A. lixula* on the Adriatic Sea and *P. lividus* on Tenerife Island, respectively. These authors found a diameter of sexual maturity for *A. lixula* (between 9.2 and 14 mm) and *P. lividus* (between 10.4 and 14.5 mm) lower than our results. In addition, [82] reported that the *P. lividus* collected were mature with a diameter test between 36–60 mm, and [77] reported an $L_{50}$ (first maturity) of 26.12 mm, and $L_{95}$ (mass maturity) at 40.56 mm in Tenerife Island. Our results show that the size of the first maturity in *P. lividus* in Gran Canaria Island is higher than in other localities around the world (42.82 mm for males and 46.03 mm for females). This pattern may be explained by reduced fishing pressure in the area, as this species is not commonly used as bait in Gran Canaria. Furthermore, its tendency to inhabit small cavities likely makes it less accessible to fishers, which may contribute to the presence of larger individuals. In the case of the other species, there are no records of the size of the first maturity in the Canarian Archipelago, for that reason, this work provides new information to understand the populations in this North Atlantic region, and to set a baseline data for management plans for sea urchin coastal populations.

## 3. DNA barcoding analysis

The COI sequences obtained in our study for *P. lividus*, *A. lixula*, and *S. granularis* showed more than 95% similarity with existing sequences in GenBank. For *P. lividus*, the sequences had between 99.54% and 99.85% similarity with over 100 GenBank sequences, including those reported by [47] (Access numbers AY630792 to AY630918), confirming species identification. Similarly, *A. lixula* sequences exhibited 99.04% and 100% similarity with more than 100 GenBank sequences, including those reported by [86] (Access numbers JQ745096 to JQ745256), further validating the

identification. For *S. granularis*, the sequence reported by [87] was 96.61% similar to ours. However, the COI region analyzed in this study differed from that of [87]. Specifically, in *S. granularis* sampled from the Canary Islands, the primer COIa (5'-AGT ATA AGC GTC TGG GTA GTC-3') was used, which amplified the region from 540 bp to 683 bp. We used the primer COIe described by [47] (5'- ATA ATG ATA GGA GGR TTT GG-3'), one of the most common primers for echinoderm sequences used worldwide, which amplifies from 1 bp to 680. Therefore, only 140 bp (26% of the sequence) overlapped between our *S. granularis* COI sequences and those amplified by [87] (Access numbers AY183288 to AY183290). Our sequences represent new fragments of the COI gene sequenced for the 3 species and available in GenBank for future studies and metabarcoding efforts.

## Conclusions

Morphological measurements of sea urchins showed a significant increase in length without a proportional gain in weight, because the weight of each organism is intimately linked to its reproductive state, given that the gonads play a significant role in the variation of the animal's weight. Sea urchin species displayed oocytes at different developmental stages, indicating asynchronous ovarian development with successive batch spawner seasons. The highest reproductive peak generally occurs during the warm season for all three species examined in this study, which could coincide with increased nutrient availability in the Canarian Archipelago. Specifically, *P. lividus* was the only species that showed two reproductive seasons in San Cristobal annually. The environmental conditions at each site seem to be the main factors influencing the differences found in the reproduction of sea urchins. Furthermore, females of our three species were mature at a larger size than males, which could contribute to the greater biomass of mature females than males and to maximizing the egg-producing biomass. In addition, the accessibility of the localities and the anthropogenic pressure (e.g., pollution, fisheries) may impact population structure and size of first maturity, consequently affecting the development of these species. The information provided in this study will offer a baseline about the population biology of sea urchin species. It will help to create a plan for fishery management along the North East Atlantic region, especially in the Canarian Archipelago, considering the real value of the minimum harvestable size of the organisms, maximum catch per capita, minimizing the pressure over this resource and protecting it for future generations.

## Acknowledgments

The authors thank the Government of the Canary Islands for providing the scientific permit to collect the samples for this study. Raibel Nuñez-Gonzalez thanks the Senckenberg Museum of Natural History and Grunelius-Möllgaard Lab technicians for helping during the DNA sample analysis. Thanks to the people who helped us during the sampling campaigns. Finally, we greatly thank the anonymous reviewers, whose comments improved the quality of this manuscript.

## Author contributions

**Conceptualization:** Raibel Z. Nuñez Gonzalez.

**Data curation:** Raibel Z. Nuñez Gonzalez, Airam N. Sarmiento-Lezcano.

**Formal analysis:** Raibel Z. Nuñez Gonzalez, Airam N. Sarmiento-Lezcano, María J. Caballero.

**Investigation:** Raibel Z. Nuñez Gonzalez, Airam N. Sarmiento-Lezcano.

**Methodology:** Raibel Z. Nuñez Gonzalez, Airam N. Sarmiento-Lezcano, María J. Caballero, Ekin Tilic.

**Project administration:** Raibel Z. Nuñez Gonzalez.

**Resources:** María J. Caballero, Ekin Tilic, José Juan Castro-Hernández.

**Supervision:** María J. Caballero, Ekin Tilic, José Juan Castro-Hernández.

**Validation:** Raibel Z. Nuñez Gonzalez, María J. Caballero, Ekin Tilic, José Juan Castro-Hernández.

**Writing – original draft:** Raibel Z. Nuñez Gonzalez, Airam N. Sarmiento-Lezcano.

**Writing – review & editing:** Raibel Z. Nuñez Gonzalez, Airam N. Sarmiento-Lezcano, María J. Caballero, Ekin Tilic, José Juan Castro-Hernández.

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
