## [Decision Letter · Decision Letter 0]

PONE-D-25-16677Locality matters: variation in the reproductive cycle and population structure of subtropical sea urchins.PLOS ONE

Dear Dr. Nuñez Gonzalez,

many thanks for submitting your manuscript to PLOS One!

First of all, I want to apologize for the long waiting time. Unfortunately, the many reviewers I invited during multiple rounds declined to review the manuscript. This had nothing to do with your manuscript itself but with their current assignments and schedule.

However, I am pleased that your manuscript only requires minor edits. The only critical point is that you must make all relevant data publicly accessible. This is PLOS Policy and cannot be waived.

All the best,

Tobias

We look forward to receiving your revised manuscript.

Kind regards,

Tobias B. Grun, Ph.D.

Academic Editor

PLOS ONE

2. In the online submission form, you indicated that [The datasets produced in this study are not publicly accessible, as they are still being analyzed for forthcoming publications. However, they can be obtained from the corresponding author upon reasonable request.].

1. You may seek permission from the original copyright holder of Figure(s) [#] to publish the content specifically under the CC BY 4.0 license. 

Additional Editor Comments (if provided):

Reviewers' comments:

Reviewer's Responses to Questions

**Comments to the Author**

1. Is the manuscript technically sound, and do the data support the conclusions?

Reviewer #1: Yes

2. Has the statistical analysis been performed appropriately and rigorously? 

Reviewer #1: Yes

3. Have the authors made all data underlying the findings in their manuscript fully available?

Reviewer #1: Yes

4. Is the manuscript presented in an intelligible fashion and written in standard English?

Reviewer #1: Yes

5. Review Comments to the Author

Reviewer #1: Thank you for preparing this manuscript. The topic is very relevant to the scientific community. I have some minor recommendations. Plese see comments in the PDF. Major point: All data must be made available prior to publication in PLOS One.

6. PLOS authors have the option to publish the peer review history of their article (what does this mean? ). If published, this will include your full peer review and any attached files.

**Do you want your identity to be public for this peer review?** For information about this choice, including consent withdrawal, please see our Privacy Policy .

Reviewer #1: No

---

## [Author Response · Author response to Decision Letter 1]

10 Jun 2025

General comments

1. Please ensure that your manuscript meets PLOS ONE's style requirements, including those for file naming. The PLOS ONE style templates

I agreed. The style requirement was reviewed.

2. In the online submission form, you indicated that [The datasets produced in this study are not publicly accessible, as they are still being analyzed for forthcoming publications. However, they can be obtained from the corresponding author upon reasonable request.].

Agreed. The datasets used in this study will be published on the PANGAEA website (https://www.pangaea.de/). However, the DOI of this dataset has not been assigned yet. We expect to obtain the DOI before the article is published, so that we can include it directly in the final version of the manuscript.

The section of the ethics statement was added in lines 191-200.

4. We note that Figure 1 in your submission contain [map/satellite] images which may be copyrighted. All PLOS content is published under the Creative Commons Attribution License (CC BY 4.0), which means that the manuscript, images, and Supporting Information files will be freely available online, and any third party is permitted to access, download, copy, distribute, and use these materials in any way, even commercially, with proper attribution. For these reasons, we cannot publish previously copyrighted maps or satellite images created using proprietary data, such as Google software (Google Maps, Street View, and Earth).

Thank you for your observation regarding Figure 1. We would like to clarify that the map was created using a Geographic Information System (GIS) software, specifically QGIS (version 3.14), and it does not include any copyrighted satellite or proprietary map imagery such as those from Google Maps or Google Earth. To address this, we have included the following sentence in the main text at line 122-124: “The map was generated using QGIS [44] and bathymetric data obtained from the GEBCO (General Bathymetric Chart of the Oceans) dataset [45].”

The base layers and data used for the map are entirely from open-access sources released under permissive licenses compatible with CC BY 4.0. In particular, we used publicly available shapefiles from Natural Earth, which are free for reuse, including commercial purposes, with proper attribution.

We confirm that Figure 1 was entirely created by the authors and is original. Therefore, we believe it fully complies with PLOS’s copyright and licensing requirements.

Reviewer 1#

1) Line 43. “With no sexual dimorphism”. Please formulate more cautious. “with no known sexual dimorphism” or similar.

Agreed. The sentence was changed according to the suggestion.

2) Line 84. “Reduces genetic variability”. Please add reference.

The reference was added in line 87.

3) Line 125. “each sea urchin species”. Please add a table showing how many sea urchin from which site were collected at what date and which size they had.

This information is shown in Table 1. The 677 samples were collected during a year, and because of that is difficult to show this information in the one table. But all the details will be shown in the dataset published.

4) Lines 125-126. “manually collected”. Please state how they were collected (e.g., SCUBA, snorkeling, etc).

Agreed.

5) Line 136. “in this study”. Please describe what happened to the samples, where they are stored and where they remain including catalogue IDs.

Agreed. The information was added in line 143.

6) Lines 152-153. “Samples were embedded in paraffin, sectioned at 4um, and stained with haematoxylin and eosin (H&E). Please describe the process more detailed or cite a protocol.

Agreed. The information was improved in lines 161-164.

7) Line 202. “R Studio software”. Please also cite used packages and libraries. Please do this for all analyses mentioned.

Agreed. The suggestion of reviewer was added in the main text (Line 202-254). All the statistical analysis was performed using R Studio software. The following R packages were used to conduct the analyses: stats for the Chi-square test, Fisher’s exact test, ANOVA, and Tukey’s HSD post-hoc test; ggplot2 for data visualization; dplyr and tidyr for data manipulation; smart for inferences of allometric measures; and car for testing model assumptions such as homogeneity of variances.

8) Line 243. “(p<0.05)”. Please use small 'p' instead of 'P'. Change all.

Agreed.

9) Line 537. “In summary”. Remove.

Agreed.

10) Line 545. Change 2 by two.

Agreed.

11) Figure 2. Please use colors that are more accessible for people with seeing impairment. Make sure that these figures can also be read in gray-scale prints. Do this for all figures.

Agreed. All the figures were improved using accessible colours for people with seeing impairment

While revising your submission, please upload your figure files to the Preflight Analysis and Conversion Engine (PACE) digital diagnostic tool, https://pacev2.apexcovantage.com/. PACE helps ensure that figures meet PLOS requirements.

Agreed. As suggested in the review, the images were optimized using PACE.

---

## [Editor Report · Decision Letter 1]

Locality matters: variation in the reproductive cycle and population structure of subtropical sea urchins.

PONE-D-25-16677R1

Dear Dr. Nuñez Gonzalez,

We’re pleased to inform you that your manuscript has been judged scientifically suitable for publication and will be formally accepted for publication once it meets all outstanding technical requirements.

Kind regards,

Tobias B. Grun, Ph.D.

Academic Editor

PLOS ONE